# Prevalence of malaria and helminth infections in rural communities in northern Sierra Leone, a baseline study to inform Ebola vaccine study protocols

Frank Baiden[1]*, Suzanne Fleck[1], Bailah Leigh[2], Philip Ayieko[1,3], Daniel Tindanbil[1], Tuda Otieno[1], Bolarinde Lawal[1], Mattu Tehtor[2], Maariam Rogers[2], Lazarus Odeny[4], Mary H. Hodges[5], Mustapha Sonnie[5], Mohamed Samai[2], David Ishola[1], Brett Lowe[1], Deborah Watson-Jones[1,3], Brian Greenwood[1]

1 London School of Hygiene & Tropical Medicine, London, United Kingdom, 2 College of Medicine and Allied Health Sciences, University of Sierra Leone, Freetown, Sierra Leone, 3 Mwanza Intervention Trials Unit, National Institute for Medical Research, Mwanza, Tanzania, 4 Kenya Medical Research Institute, Centre for Respiratory Diseases Research, Nairobi, Kenya, 5 Helen Keller International, Freetown, Sierra Leone

* frank.baiden@lshtm.ac.uk

**Data Availability Statement:** All relevant data are within the paper and its Supporting Information files.

## Abstract

### Introduction

Recurrent parasitic infections may influence the immune response to vaccines. In the Partnership for Research on Ebola VACcinations extended follow-UP and clinical research capacity build-UP (PREVAC-UP) study being undertaken in Mambolo, northern Sierra Leone, participants are being followed up to assess the potential impact of exposure to malaria and/or helminth infections on long-term immune response to two Ebola vaccines. To support the development of the assays that will be used in this evaluation, a parasitological survey was conducted in Mambolo between November 2019 and February 2020.

### Methods

Healthy individuals aged ≥1 year who were resident in Mambolo Chiefdom were selected using a stratified sampling approach and questionnaires were administered to explore their sociodemographic characteristics. Microscopy was used to detect malaria parasites, intestinal helminths and urinary schistosome infections. Rapid blood tests were used to detect infections with *Onchocerca volvulus* and *Wuchereria bancrofti*. We estimated the overall prevalence of these infections and used adjusted logistic regression models to explore risk factors for malaria and hookworm infection.

### Results

Eight hundred and fifteen (815) residents, 50.9% of whom were female were surveyed. Overall, 309 (39.1%) of 791 persons tested for malaria had a positive blood slide; *Plasmodium falciparum* was the dominant species. Helminth infection was detected in 122 (15.0%) of 815 stool samples including three mixed infections. The helminth infections comprised

**Funding:** This project is part of the EDCTP2 programme supported by the European Union (grant number RIA2017S-2014 – PREVAC-UP) and by the US National Institute of Allergy and Infectious Diseases of the National Institutes of Health. This research was supported in part by the National Institutes of Health (NIH), by Institut national de la santé et de la recherche médicale (Inserm) and by the London School of Hygiene and Tropical Medicine (LSHTM). There are no grant numbers for the funding from the US NIAID, NIH, INSERM or LSHTM The funders had no role in study design, data collection and analysis, decision to publish, or preparation of the manuscript. FB, SF, BL, PA, DT, TO, BL, MT, MR, LO, MS, DI, DWJ & BG received full or partial salary support from the grant 'RIA2017S-2014 – PREVAC-UP' through LSHTM. The authors did not receive specific funding for this work from any other funder.

**Competing interests:** The authors have declared that no competing interests exist.

102 (12.5%) cases of hookworm, 11 (1.3%) cases of *Trichuris trichiura*, 10 (1.2%) cases of *Schistosoma mansoni* and two (0.2%) cases of *Ascaris lumbricoides*. Being male (OR = 2.01, 95% CI 1.15–3.50) and residing in a non-riverine community (OR = 4.02, 95%CI 2.32–6.98) were the factors associated with hookworm infection. *Onchocerca volvulus* and *Wuchereria bancrofti* infections were found in 3.3% and 0.4% of participants respectively.

## Conclusion

Malaria and hookworm are the most prevalent parasite infections and those most likely to influence long-term immune response to Ebola vaccines among the trial participants.

## Introduction

Sierra Leone has an estimated population of eight million and an under-five mortality rate of 105 per 1,000 live births [1]. The country was severely affected by the 2014–2016 West African Ebola epidemic which resulted in 28,616 cases and 11,310 deaths in Guinea, Liberia and Sierra Leone [2]. In 2018, the London School of Hygiene and Tropical Medicine (LSHTM) in collaboration with the College of Medicine and Allied Health Sciences (COMAHS), University of Sierra Leone, the United States National Institute of Health (NIH), the French National Institute of Health and Medical Research (INSERM) and other partners initiated the Ebola vaccine study (PREVAC) in the Kambia district in northwest Sierra Leone.

This trial is investigating the initial and long-term safety and immunogenicity of the two prophylactic Ebola vaccines, the two-dose heterologous vaccine regimen of Ad26.ZEBOV and MVA-BN-Filo vaccines (Janssen Vaccines & Prevention B.V.) and a single or boosted regimen of the recombinant Vesicular Stomatitis Virus (rVSV) vaccine (Merck Sharp and Dohme) (https://clinicaltrials.gov/ct2/show/NCT02876328) [3]. Initial studies of the vaccine regimens have demonstrated the safety and immunogenicity of each vaccine [3–5]. The rVSV vaccine has shown protection against Ebola disease using ring vaccination in Guinea and the Democratic Republic of Congo (DRC) [6, 7]. The duration of protective levels of antibody to each vaccine regimen is being evaluated in an extension to the PREVAC clinical trial (the PREVAC-UP study) [8]. The Ebola vaccines are evaluated in persons aged one year and older.

There is evidence that malaria and some helminth infections can suppress the immediate response to some vaccines [9–14]. However, there is little information as to whether repeated exposure to malaria and helminth infections influences the rate in decline in antibody titres, and consequently the time when a booster dose of vaccine might be needed in exposed populations. Determining the potential impact of repeated exposure to malaria and/or to helminths on the duration of the immune response to three Ebola vaccine regimens is one of the objectives of the PREVAC-UP study. This will be done through measuring antibody titres to malaria and to selected helminths at annual surveys for five years after vaccination using a Luminex platform [15]. In order to determine which antigens to employ in the Luminex assay, a cross-sectional parasitological survey has been conducted in the community where the PREVAC-UP survey is being undertaken and the results of this survey are presented in this paper.

## Methods

### Study site

The study was conducted in Mambolo Chiefdom on the southern border of Kambia district, northern Sierra Leone. The Chiefdom covers an area of 230km². It is largely rural and has an

estimated population of 37,952; approximately 36% of inhabitants are children aged below 11 years of age [16]. Most residents of Mambolo town are subsistence farmers who cultivate rice and vegetables in swampy areas along the Great Scarcies River. This river, which is fast flowing in some areas, is a source of water for domestic use, irrigation, freshwater fishing and recreational swimming. Other sources of water in the Chiefdom include boreholes. The residents of Malambay and Robana villages, situated about 10 kilometres away from the river, live on drier land and do not have as much access to the river. The Chiefdom is thus distinctly divided into riverine and non-riverine areas.

The Mambolo health centre (MHC) is the main health facility in the Chiefdom. It is managed by a community health officer and community health nurses and offers primary health care that include maternal and child health services. It also oversees the sub-district implementation of malaria control interventions such as the distribution of insecticide-treated nets (ITN) and Mass Drug Administration (MDA) in the control of Soil-Transmitted Helminths (STH), and other Neglected Tropical Diseases (NTDs): Lymphapatic filariasis (LF) and onchocerciasis (OV). Mass distribution of ITNs at a rate of one per 3 individuals per household is undertaken every four years. Pregnant women and children aged less than five years are routinely given ITNs at antenatal and child health clinics respectively.

Deworming of School-Aged Children (SAC) with albendazole has been part of the public health system in Sierra Leone for many years. The program defined SAC to be children aged between 5 and 14 years. A national impact survey in 2016 however demonstrated that the intervention was no longer justified. MDA-LF with ivermectin and with albendazole was undertaken annually from 2008–2017 for everyone over 5 years of age. Since 2018 however, this activity has been targeted at only SAC. Deworming of pre-SAC with albendazole continues to be undertaken every six months. This is done through local health facilities and community outreach services.

## Selection of houses and participants for the parasitological survey

All houses in Mambolo town (riverine) and Malambay and Robana villages (non-riverine) were identified, mapped, and numbered. The target population size for the survey was 500 participants each in riverine and non-riverine communities. Assuming an average number of eligible (aged 1 year or older) persons to be five per household, probability proportional to size was used to determine the number of houses that needed to be visited to achieve the target number of participants. The houses were selected randomly.

## Procedures

The cross-sectional survey was conducted between November 2019 and February 2020. This period was within the dry season when cases of malaria are generally expected to be lower than would be the case in the rainy season. Trained fieldworkers visited each selected house and identified all members of the household who were aged one year or older. After informed consent had been obtained, fieldworkers administered a paper-based questionnaire to each head of household to collect data on the demographic characteristics of members of the household and on the household's characteristics.

Each enumerated member of household, or their guardian if they were aged less than 18 years, was interviewed with a paper-based questionnaire to obtain data on educational status, occupation, travel history, recent illnesses, and vaccination status (young children). Data were also collected on health-seeking behaviours, experience of fever, infection prevention practices, such as possession and use of an ITN, and uptake of MDA. The interviews were conducted in local languages. The questionnaires are attached as supplements to this paper.

## Sample collection

Trained phlebotomists took 2.5mls of venous blood into appropriately labelled collection tubes. Participants were given pre-labelled containers and information on how to provide urine and stool samples under hygienic conditions. All samples were analysed according to standard operating procedures at the research laboratory in Kambia town.

Thick and thin blood smears were Giemsa-stained and read by two experienced microscopists. Slides were declared negative only after 100 high power fields had been examined. When a slide was positive, parasites were counted against 200 white blood cells to determine parasite density. Discordant slides from the two microscopists in either positivity or density according to a standardise algorithm were read by a third microscopist [17]. The concordant results for parasitaemia and species, and the average of the two closest parasite densities were taken as the final results.

Rapid diagnostic tests for *Onchocerca volvulus* (O. *volvulus*) and *Wuchereria bancrofti* (*W. bancrofti*) were performed with SD Bioline Onchocerciasis IgG4 Rapid Test (Alere, Standard Diagnostics, Inc, Yongin-si, Republic of Korea) and Alere Filariasis Test Strips (Alere Scarborough, USA) respectively. These rapid diagnostic test kits qualitatively detect IgG4 antibodies against the Ov16 antigen and circulating *W. bancrofti* antigens respectively.

The presence of *Schistosoma haematobium* eggs in urine was explored using the Schistosome Test Kit (Sterlitech Corporation, Kent, USA). Kato-Katz kits (Sterlitech Corporation, Kent, USA) were used to prepare slides from stool samples that were examined for eggs of *S. mansoni* and other intestinal helminths. As part of quality control for the procedure, the microscopic examination took place within 30 minutes after sample processing. Ten percent of negative samples and all positive samples were reread by a senior laboratory staff immediately after the first reading to validate the results.

## Data management and analysis

Completed questionnaires and laboratory results were double entered using a platform created in OpenClinica Community Version 3.2. Clean data were exported and analysed with STATA version 15.1.

Descriptive data analysis was done for all survey participants. The overall prevalence of each parasitic infection (malaria, intestinal helminths, *O. volvulus* and *W. bancrofti*) was calculated for all participants then further analysed to obtain prevalence for riverine and non-riverine areas separately. We used sampling weights to account for the sampling design in the regression analyses. The regression analyses were used to explore risk factors for malaria and hookworm infections. The number of positive samples for the remaining parasitic infections was too small for risk factors to be determined. Odds ratios and 95% confidence intervals were obtained for each explanatory variable and malaria infection or hookworm infection in the respective logistic models. Wald tests, adjusted for sampling design were used in determining statistical significance. The association between age, sex and parasitic infections were not adjusted for other explanatory variables as age and sex were considered *a priori* potential confounders for the association between malaria infection and area of residence, sleeping under an ITN and duration since last fever episode. Similarly, in the logistic regression for hookworm infection, age and sex were *a priori* confounders of the association between infection and area of residence, time since last MDA, and frequency of contact with water from river, lake, canal or rice field. The variables with a p value <0.05 were considered to be significantly associated with malaria or hookworm infection depending on the appropriate model. This model building strategy, that takes the interrelationship between explanatory variables into consideration,

ensures that the effects of both distal and proximal determinants of parasitic infection are estimated appropriately.

### Ethical considerations

The study was approved by the ethics committees of LSHTM and the Sierra Leone Ethics and Scientific Review Committee of the Ministry of Health and Sanitation. Community consent was obtained through meetings with the paramount chief and community elders. Meetings were organised in the communities to inform people about the survey and the proposed procedures. Written informed consent was obtained from the heads of each selected households and individual participants. Written assent was obtained from household members who were aged 12–17 years (inclusive) while a parent or guardian gave consent on behalf of children who were 12 years of age or younger. When a participant could not read or write in English, consent and or assent was obtained with the help of an impartial witness. Each participant was given 15,000 Sierra Leone Leones (approximately USD 1.50) as compensation for the time spent in participating in the survey.

Participants who had fever or other symptoms suggestive of malaria within three days of the survey were tested for malaria using *Abbott* SD BIOLINE Malaria Ag P.f/Pan rapid diagnostic test and immediately treated with artemether and lumefantrine if this was positive. Other reported minor ailments were treated by nurses who were part of the survey team. After laboratory examination of stool samples, participants who had intestinal helminth infections, including *S. mansoni*, were offered treatment according to standard practice of praziquantel (using a dose pole) [18, 19] or albendazole 200mg for children 1–2 years of age and 400mg for those two years and older.

## Results

Two hundred household heads were invited to participate in the survey, 165 of whom (82.5%) agreed to participate. The overall response rate was 815/992. The response rate in riverine communities was significantly higher at 505/549 (92%) versus 310/443 (70%, p < 0.001) in non-riverine communities. Overall, the median number of household members living in a household was 4 (range 1 to 16). The median number of household members in riverine communities was 5 (1–16) versus 3 (1 to 13) in non-riverine communities. Of the 815 participants, 415 (50.9%) were female and 402/813 (49.4%) were aged 18 years or older (**Table 1**) and 772 out of 813 (95.0%) were Themne. Most participants were Muslim (798/ 815, 97.9%) and the predominant occupation was farming (205/ 815, 25.2%). (**Table 1**).

### Malaria parasitaemia

Of 815 participants, 791 (97.1%) had malaria microscopy results available; 309/791 (39.1%) participants had a positive malaria slide. A single malaria parasite species was identified in 288 slides, two species were identified in 13 slides (11 involved mixed infections of *P. falciparum* and *P. malariae* while two involved mixed infections of *P. falciparum* and *P. ovale*) and speciation was not reported for 8 slides. Thus, there were 324 positive speciation results from 301 out of 309 malaria positive samples; 287/791 (36.3%) samples were positive for *P. falciparum*, whilst *P.malariae*, *P.ovale and P.vivax* were identified in 29/791 (3.7%), 7/791 (0.9%), and 1/ 791 (0.1%) samples, respectively (**Table 2**).

The proportion of individuals with malaria parasitaemia was similar among participants who were resident in riverine (180/488; 36.9%) or non-riverine communities (129/303; 42.6%). Similarly, the prevalence of malaria parasitaemia was similar in males and females (39.6% [153/386] versus 38.9% [156/401], respectively). Twenty-one (26.9%) out of the 78

**Table 1. Characteristics of participants in the parasitology survey in the Mambolo Chiefdom.**

| | n/N | Percent (%) |
|---|---|---|
| **Age group of participants** | | |
| 1–4 years | 82/813 | 10.1% |
| 5–11 years | 189/813 | 23.2% |
| 12–17 years | 140/813 | 17.2% |
| > = 18 years | 402/813 | 49.4% |
| **Sex** | | |
| Female | 415/815 | 50.9% |
| Male | 400/815 | 49.1% |
| **Ethnic group** | | |
| Themne | 772/813 | 95.0% |
| Soso | 17/813 | 2.1% |
| Limba | 18/813 | 2.2% |
| Others | 6/813 | 0.7% |
| **Occupation** | | |
| Farmer | 205/813 | 25.2% |
| Fisherman | 63/813 | 7.7% |
| Petty trader | 40/813 | 4.9% |
| Formal employment | 37/813 | 4.5% |
| Student | 468/813 | 57.6% |
| **Marital status** | | |
| Single | 256/544 | 47.1% |
| Married | 254/544 | 46.7% |
| Divorced/Separated | 8/544 | 1.5% |
| Widowed | 26/544 | 4.8% |
| **Religion** | | |
| Christian | 17/815 | 2.1% |
| Muslim | 798/815 | 97.9% |
| **Area of residence** | | |
| Riverine | 505/815 | 62.0% |
| Non-riverine | 310/815 | 38.0% |

participants aged 1–4 years had malaria parasitaemia compared to 91/183 (49.7%), 87/135 (64.4%), and 110/393 (28.0%) among participants aged 5–11 years, 12–17 years and 18 years and above, respectively. Compared to participants aged 5-11years, participants aged 1–4 years (Odds Ratio = 0.35, 95% CI 0.20–0.62) and 18 years and older (OR = 0.41, 95% CI 0.26–0.66) were less likely to have malaria parasitaemia. Participants who were aged 12-17yrs (OR = 1.94, 95% CI 1.16–3.24) were more likely to have malaria parasitaemia than those aged 5–11 years. In the adjusted analysis, age and time since last episode of fever were associated with malaria parasitaemia. The participants who did not sleep under an ITN were more likely to have malaria parasites than those who did (43.6% versus 38.3%; OR = 1.63, 95%CI 0.97–2.76) (**Table 3**).

**Soil-transmitted helminths and schistosome infections** There were 125 positive tests for helminth infection and these positive tests occurred in 122 participants (three stool samples had mixed infections accounting for 6 infections). **The positive helminth infections comprised 102 (12.5%) cases of hookworm, 11 (1.3%) cases of *T. trichiura*, 10 (1.2%) cases of *S. mansoni* and two (0.2%) cases of *A. lumbricoides*. The three stool samples showing mixed infections were two samples with hookworm and *T. trichiura* mixed infection, and one**

**Table 2. Prevalence of parasitic infections in participants in Mambolo Chiefdom.**

|  | Riverine | Non-riverine | Total |
|---|---|---|---|
| Malaria infection* | | | |
| *P. falciparum* | 164/488 (33.6%) | 123/303 (40.6%) | 287/791 (36.3%) |
| *P. malariae* | 23/488 (4.7%) | 6/303 (2.0%) | 29/791 (3.7%) |
| *P. ovale* | 4/488 (0.8%) | 3/303 (1.0%) | 7/791 (0.9%) |
| *P. vivax* | 1/488 (0.2%) | 0/303 (0.0%) | 1/791 (0.1%) |
| Stool samples for intestinal helminths | | | |
| Hookworm | 28/505 (5.5%) | 74/310 (23.9%) | 102/815 (12.5%) |
| *A. lumbricoides* | 2/505 (0.4%) | 0/310 (0.0%) | 2/815 (0.2%) |
| *T. trichiuria* | 6/505 (1.2%) | 5/310 (1.6%) | 11/815 (1.3%) |
| *S. mansoni* | 9/505 (1.8%) | 1/310 (0.3%) | 10/815 (1.2%) |
| Urine for *S. heamatobium* | 0/505 (0.0%) | 0/310 (0.0%) | 0/815 (0.0%) |
| Filariae rapid blood tests | | | |
| *O. volvulus*[†] | 13/503 (2.6%) | 14/309 (4.5%) | 27/812 (3.3%) |
| *W. bancrofti*[†] | 2/503 (0.4%) | 1/309 (0.3%) | 3/812 (0.4%) |

*[†]Differences in denominators are due to either incomplete laboratory analyses or non-provision of samples by participants

**with hookworm and schistosomiasis mixed infection** (Table 2). Fifty (43.5%) out of the 122 cases with evidence of helminth infection were female. The median age of participants with positive stool egg findings was 17.5 (interquartile range of 10–35) years. Thirty-seven (32.2%) cases were in children aged less than 12 years of age. No *S. haematobium* eggs were detected on urine microscopy.

Compared with participants who lived in the riverine communities, participants who resided in the non-riverine communities were about four times more likely to have hookworm infection (AOR = 4.02, 95% CI 2.32–6.98). (Table 4) Participants who made weekly, monthly or rare contacts with the river or rice fields and those who made daily contact were more likely to have hookworm infection than those who did not have any contact with the river or rice field. (AOR = 2.03, 95% CI 1.03–3.99 and AOR = 3.34, 95% CI 1.79–6.23 respectively).

Compared to participants aged 5-11years, those aged 1–4 years were less likely to have hookworm infection (AOR = 0.06, 95% CI 0.01–0.47). Males were more likely to have hookworm infection than female participants (AOR = 2.01, 95% CI 1.15–3.50). Participants who had never taken albendazole were less likely to have hookworm infection compared with participants who had taken albendazole as part of an MDA within six months prior to the survey, (AOR = 0.25, 95%CI 0.07–0.83) (Table 4).

## Blood associated parasites (O. volvulus and W. bancrofti)

Samples from 27/812 (3.3%) participants tested positive for *O. volvulus* using a rapid serology test. Only three of these 27 participants were aged below 18 years (two 10-year-olds and one 11-year-old). Only three of 812 (0.4%) samples tested were positive based on a rapid serology test for *W. bancrofti* test (Table 2). The three cases were aged 17, 27 and 46 years old; two were females. One participant with a positive test for *O. volvulus* had travelled to the Republic of Guinea in February 2019 where he stayed for three months. None of the remaining 29 participants with positive test for *O. volvulus* or *W. bancrofti* had travelled out of Sierra Leone within the 12 months preceding the survey.

**Table 3. Risk factors for malaria infection among participants in Mambolo Chiefdom.**

| | | Malaria | | OR (95% CI) | P value | AOR (95% CI) | P value |
|---|---|---|---|---|---|---|---|
| | | Yes | No | | | | |
| **Age group** | | | | | | | |
| 1–4 years | | 21 (26.9) | 57 (73.1) | 0.33 (0.20–0.56) | <0.001 | 0.35 (0.2–0.62) | <0.001 |
| 5–11 years | | 91 (49.7) | 92 (50.3) | 1.00 (Ref) | | 1.00 (Ref) | |
| 12–17 years | | 87 (64.4) | 48 (35.6) | 1.86 (1.14–3.05) | 0.014 | 1.94 (1.16–3.24) | 0.011 |
| > = 18 years | | 110 (28.0) | 283 (72.0) | 0.41 (0.28–0.61) | <0.001 | 0.41 (0.26–0.66) | <0.001 |
| **Sex** | | | | | | | |
| Female | | 156 (38.9) | 245 (61.1) | 1.00 (Ref) | | 1.00 (Ref) | |
| Male | | 153 (39.6) | 233 (60.4) | 1.07 (0.76–1.50) | 0.695 | 1.09 (0.78–1.52) | 0.607 |
| **Ethnic group** | | | | | | | |
| Themne | | 291 (38.9) | 458 (61.1) | 1.00 (Ref) | | | |
| Soso | | 7 (41.2) | 10 (58.8) | 1.09 (0.37–3.20) | 0.877 | | |
| Limba | | 7 (41.2) | 10 (58.8) | 1.19 (0.53–2.67) | 0.679 | | |
| **Occupation** | | | | | | | |
| Farmer | | 62 (30.8) | 139(69.2) | 1.00(1.00–1.00) | | | |
| Fisherman | | 19 (30.2) | 44(69.8) | 0.95(0.48–1.88) | 0.878 | | |
| Petty trader | | 11 (28.9) | 27(71.1) | 1.28(0.61–2.68) | 0.51 | | |
| Formal employment | | 8 (21.6) | 29(78.4) | 0.69(0.25–1.92) | 0.477 | | |
| Student | | 209 (46.4) | 241(53.6) | 2.02(1.27–3.22) | 0.003 | | |
| Not reported | | 0 (0.0) | 2(100.0) | 1.00(1.00–1.00) | | | |
| **Marital status** | | | | | | | |
| Single | | 123 (49.4) | 126 (50.6) | 1.00 (Ref) | | | |
| Married | | 66 (26.7) | 181 (73.3) | 0.39 (0.23–0.66) | <0.001 | | |
| Divorced/Separated | | 1 (12.5) | 7 (87.5) | 0.03 (0.00–0.25) | 0.002 | | |
| Widowed | | 7 (26.9) | 19 (73.1) | 0.32 (0.11–0.93) | 0.036 | | |
| **Area of residence** | | | | | | | |
| Riverine | | 180 (36.9) | 308 (63.1) | 1.00 (Ref) | | 1.00 (Ref) | |
| Non-riverine | | 129 (42.6) | 174 (57.4) | 1.46 (0.93–2.29) | 0.098 | 1.53 (0.94–2.5) | 0.089 |
| **Slept under ITN last night** | | | | | | | |
| | Yes | 258 (38.3) | 416 (61.7) | 1.00 (Ref) | | 1.00 (Ref) | |
| | No | 51 (43.6) | 66 (56.4) | 1.27 (0.80–2.01) | 0.306 | 1.63 (0.97–2.76) | 0.066 |
| **Number of nights that participant slept under ITN over the last 3 days** | | | | | | | |
| None | | 40 (39.6) | 61 (60.4) | 0.95 (0.53–1.68) | 0.847 | | |
| 1 night | | 3 (25.0) | 9 (75.0) | 0.71 (0.17–3.00) | 0.644 | | |
| 2 nights | | 13 (35.1) | 24 (64.9) | 0.65 (0.28–1.49) | 0.308 | | |
| 3 nights | | 253 (39.5) | 388 (60.5) | 1.00 (Ref) | | | |
| **At least one ITN at home** | | | | | | | |
| | Yes | 183 (37.3) | 308 (62.7) | 1.00 (Ref) | | | |
| | No | 11 (35.5) | 20 (64.5) | 0.89 (0.40–1.98) | 0.771 | | |
| **Enrolled in PREVAC study** | | | | | | | |
| | Yes | 32 (50.0) | 32 (50.0) | 1.00 (Ref) | | | |
| | No | 274 (38.0) | 447 (62.0) | 0.68 (0.35–1.33) | 0.26 | | |
| **Duration since last fever episode Please clarify = 6 months, ≤ or ≥** | | | | | | | |
| < 6 months | | 269 (39.6) | 411 (60.4) | 1.00 (Ref) | | | |
| > 6 months | | 15 (26.3) | 42 (73.7) | 0.45 (0.22–0.90) | 0.025 | 0.38 (0.18–0.79) | 0.01 |

**Table 4. Risk factors for hookworm infection in Mambolo Chiefdom.**

| | Hookworm | | OR (95% CI) | P value | AOR(95% CI) | P value |
|---|---|---|---|---|---|---|
| | **Yes** | **No** | | | | |
| **Age group** | | | | | | |
| 1–4 years | 1 (1.2) | 81 (98.8) | 0.05 (0.01–0.38) | 0.004 | 0.06 (0.01–0.47) | 0.008 |
| 5–11 years | 26 (13.8) | 163 (86.2) | 1.00 (Ref) | | 1.00 (Ref) | |
| 12–17 years | 20 (14.3) | 120 (85.7) | 1.07 (0.58–1.97) | 0.822 | 0.92 (0.5–1.7) | 0.787 |
| > = 18 years | 54 (13.4) | 348 (86.6) | 0.90 (0.48–1.68) | 0.732 | 0.7 (0.36–1.36) | 0.295 |
| **Sex** | | | | | | |
| Female | 38 (9.2) | 375 (90.8) | 1.00 (Ref) | | 1.00 (Ref) | |
| Male | 64 (16.1) | 334 (83.9) | 2.23 (1.33–3.73) | 0.003 | 2.01 (1.15–3.50) | 0.014 |
| **Ethnic group** | | | | | | |
| Themne | 93 (12.0) | 679 (88.0) | 1.00 (Ref) | | | |
| Soso | 3 (17.6) | 14 (82.4) | 2.31 (0.56–9.55) | 0.247 | - | - |
| Limba | 5 (27.8) | 13 (72.2) | 3.70 (1.47–9.32) | 0.006 | | |
| **Occupation** | | | | | | |
| Farmer | 40(19.5) | 165(80.5) | 1.00(Ref) | | | |
| Fisherman | 6(9.5) | 57(90.5) | 0.42(0.17–1.04) | 0.061 | - | - |
| Petty trader | 0(0.0) | 40(100.0) | 1.00(1.00–1.00) | | | |
| Formal employment | 3(8.1) | 34(91.9) | 0.47(0.15–1.45) | 0.188 | | |
| Student | 53(11.3) | 415(88.7) | 0.60(0.34–1.08) | 0.089 | | |
| Not reported | 0(0.0) | 2(100.0) | 1.00(1.00–1.00) | | | |
| **Marital status** | | | | | | |
| Single | 37 (14.5) | 219 (85.5) | 1.00 (Ref) | | | |
| Married | 33 (13.0) | 221 (87.0) | 0.73 (0.39–1.39) | 0.337 | - | - |
| Divorced/ Separated | 1 (12.5) | 7 (87.5) | 1.09 (0.12–9.77) | 0.935 | | |
| Widowed | 4 (15.4) | 22 (84.6) | 0.55 (0.17–1.80) | 0.319 | | |
| **Area of residence** | | | | | | |
| Riverine | 28 (5.5) | 477 (94.5) | 1.00 (Ref) | | 1.00 (Ref) | |
| Non-riverine | 74 (23.9) | 236 (76.1) | 4.14 (2.37–7.22) | <0.001 | 4.02 (2.32–6.98) | <0.001 |
| **Enrolled in PREVAC** | | | | | | |
| Yes | 2 (3.1) | 63 (96.9) | 1.00 (Ref) | | - | - |
| No | 100 (13.5) | 643 (86.5) | 5.18 (1.24–21.71) | 0.025 | | |
| **Contact with water from a river/lake/canal/rice field** | | | | | | |
| Not at all | 26 (7.3) | 330 (92.7) | 1.00 (Ref) | | 1.00 (Ref) | |
| Weekly/ monthly/rarely | 50 (14.7) | 290 (85.3) | 2.49 (1.24–4.97) | 0.01 | 2.03 (1.03–3.99) | 0.04 |
| Daily | 26 (22.4) | 90 (77.6) | 4.08 (2.10–7.91) | <0.001 | 3.34 (1.79–6.23) | <0.001 |
| **Last time you took tablet during MDA** | | | | | | |
| Never been given | 5 (6.3) | 75 (93.8) | 0.22 (0.07–0.66) | 0.007 | 0.25 (0.07–0.83) | 0.024 |
| < 6 months | 60 (13.3) | 390 (86.7) | 1.00 (Ref) | | 1.00 (Ref) | |
| 6–12 months ago | 15 (10.8) | 124 (89.2) | 0.67 (0.34–1.29) | 0.224 | 0.6 (0.32–1.14) | 0.116 |
| > 12months ago | 22 (15.2) | 123 (84.8) | 1.04 (0.60–1.83) | 0.882 | 1.09 (0.56–2.1) | 0.800 |

## Discussion

This survey, conducted in a rural community in northern Sierra Leone, was undertaken as part of a larger study to explore how repeated parasite infections could influence immune responses to the two currently licenced vaccines against EVD. Specifically, the findings will assist in defining the antigens that will be included in a Luminex assay that will be used to detect antibodies to parasite infections in this community. Response to participation in the

survey was better in the riverine than non-riverine communities. The riverine communities are closer to the offices of the Ebola vaccine project. It is conceivable that community engagements activities in the Ebola vaccine project may have impacted positively on the willingness of people in the riverine communities to participate in the survey compared to people in the non-riverine communities.

The findings of the survey demonstrated a continuing high malaria burden in this part of Sierra Leone but a low prevalence of helminth infections. The prevalence of hookworm infection was comparatively higher at 12.5%. Based on the findings of the survey, malaria and hookworm antigens are strong candidates for incorporating in the Luminex assay.

There is evidence of a substantial reduction in malaria transmission in many parts of sub-Saharan Africa in the past two decades although this has stalled in recent years [20]. However, the prevalence of malaria in Sierra Leone remains high. During the 2016 Sierra Leone Malaria Indicator Survey (SLMS), Kambia and Port Loko were among districts with the highest levels of malaria parasitaemia among children aged less than five years (48% and 59% respectively) [21]. The prevalence in Port Loko District was the highest in the country. The Mambolo Chiefdom, although geographically defined within Kambia district, lies on the boundary of Port Loko district. It is therefore not surprising that, in the present survey, a high prevalence of malaria parasitaemia was found, with age being an important determinant of asymptomatic infection. The prevalence of malaria in children aged 1–4 years in our study (26.9%) is lower than the prevalence of 39.5% found in the 2016 SLMIS for children aged 1–4 years. The higher prevalence in the 2016 SLMIS may have been due to the use of RDTs to detect diagnosis malaria parasitaemia in that survey. The trend in malaria prevalence across the age groups i.e., lower in under-five year old children and adults, but higher in older children and adolescents, is consistent with the pattern now seen in many sub-Saharan African communities, for example in community surveys in Uganda and The Gambia [22, 23]. This may be due in part to the fact that older children do not use ITNs as much as under-five year old children and adults.

The findings from the current study highlight the need to better understand the factors that sustain continuing high levels of malaria parasitaemia in the healthy population in Mambolo and similar areas and how this might be reduced. Residents of Mambolo district have good access to diagnosis and treatment, supported in part by the Ebola vaccine research initiatives, and a relatively high level of use of ITNs. Thus, although improvements could be achieved by increasing coverage with existing interventions, novel interventions are needed for communities such as Mambolo if the prevalence of malaria is to be reduced substantially.

In contrast to malaria, the survey found that the prevalence of lymphatic filariasis and onchocerciasis in Mambolo was very low. The National Neglected Tropical Diseases program (NNTDP) has benefitted from over a decade of funding from USAID and technical support from Helen Keller International. Although this study did not employ the WHO recommended methodology for conducting Transmission Assessment Survey (TAS) [24, 25] the findings align well with previous TASs conducted by the NNTDP where the prevalence of *W. bancrofti* was below the critical cut-off value for sustaining transmission. This survey results also suggest that the prevalence of *O. vulvuus* is also approaching a level at which transmission will no longer be sustainable [18, 19].

Schistosomiasis is targeted for control by the NNTDP using preventive chemotherapy. The World Health Organization (WHO) recommends that all SAC and high-risk adults (occupationally exposed to snail infested fresh water) should receive praziquantel annually/biennially in communities that are highly/moderately endemic (<50%, 10–50% prevalence). A 2016 national impact survey found the prevalence of *S. mansoni* and *S. haematobium* in Kambia to be 0.4% and 0.4% respectively. The low prevalence is most likely to be due to the lack of the

intermediate hosts (fresh-water snails) in the brackish waters of low-altitude coastal districts [26].

The NNTDP also aimed to control the transmission of STH MDA delivered according to WHO protocols for high, moderate and low risk prevalence by age groups. MDA for STH has targeted pre-school aged children twice annually since 2006. Annual MDA for LF with Ivermectin and Albendazole included SAC and adults from 2008–2017. From 2018 to date annual MDA for onchocerciasis has included Albendazole for SAC. Except in the case of hookworm, these results are consistent with the decreasing trend in prevalence as observed in the NNTDP baseline survey in 2008 (any STH: 36%, hookworm 23%) and the impact survey in 2016 (any STH: 10%, hookworm 1.2%). No moderate or high intensity STH infections were found in Kambia in 2016.

## Conclusion

This study has shown that the prevalence of malaria in Mambolo, where the PREVAC-UP Ebola vaccine trial is being conducted, is still very high and it is possible that this infection could impact on the persistence and/or level of anti-Ebola antibodies following vaccination. In contrast, the prevalence of helminth infections is low. It will be important that the Luminex serological assay is able to detect antibodies against hookworm which is now the only STH of note. The findings from this survey confirm the progress that the NNTDP is making towards control and elimination of NTDs.

## Supporting information

**S1 Data.**
(XLSX)

**S1 Questionnaire.**
(DOCX)

## Acknowledgments

The authors would like to express their gratitude to the paramount chief, elders and people of the Mambolo Chiefdom for their support and participation in the survey. They acknowledge Dr. Brima Kargbo, Chief Medical Officer and the late Dr. Samuel Juana Smith of the Ministry of Health and Sanitation, Sierra Leone. The authors are also grateful to Professor Umberto D'Alessandro of MRC Unit The Gambia at LSHTM for his critical review of the survey tools.

## Author Contributions

**Conceptualization:** Frank Baiden, Suzanne Fleck, Bailah Leigh, Tuda Otieno, Lazarus Odeny, David Ishola, Brett Lowe, Deborah Watson-Jones, Brian Greenwood.

**Data curation:** Frank Baiden, Daniel Tindanbil, Tuda Otieno, Bolarinde Lawal, Mattu Tehtor, Maariam Rogers, Lazarus Odeny, Deborah Watson-Jones.

**Formal analysis:** Frank Baiden, Suzanne Fleck, Philip Ayieko, Daniel Tindanbil, Tuda Otieno, Maariam Rogers, Lazarus Odeny, Brett Lowe, Deborah Watson-Jones, Brian Greenwood.

**Funding acquisition:** Mohamed Samai, David Ishola, Deborah Watson-Jones, Brian Greenwood.

**Investigation:** Frank Baiden, Tuda Otieno.

**Methodology:** Frank Baiden, Philip Ayieko, Daniel Tindanbil, Tuda Otieno, Bolarinde Lawal, Mattu Tehtor, Mary H. Hodges, Mustapha Sonnie, Brett Lowe, Deborah Watson-Jones, Brian Greenwood.

**Project administration:** Frank Baiden, Suzanne Fleck, Bailah Leigh, Daniel Tindanbil, Tuda Otieno, Bolarinde Lawal, Mattu Tehtor, Maariam Rogers, Mohamed Samai, David Ishola, Brett Lowe, Brian Greenwood.

**Resources:** Mary H. Hodges, Mustapha Sonnie, Mohamed Samai.

**Supervision:** Mohamed Samai, Brett Lowe, Deborah Watson-Jones, Brian Greenwood.

**Validation:** Mustapha Sonnie.

**Writing – original draft:** Frank Baiden, Suzanne Fleck, Deborah Watson-Jones, Brian Greenwood.

**Writing – review & editing:** Frank Baiden, Suzanne Fleck, Bailah Leigh, Philip Ayieko, Daniel Tindanbil, Mattu Tehtor, Lazarus Odeny, Mary H. Hodges, David Ishola, Brett Lowe, Deborah Watson-Jones, Brian Greenwood.

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
