## [Decision Letter · Decision Letter 0]

28 Feb 2022

PONE-D-21-38636Prevalence of malaria and helminth infections in rural communities in northern Sierra Leone, a baseline study to inform Ebola vaccine study protocolsPLOS ONE

Dear Dr. Baiden,

Thank you for submitting your manuscript to PLOS ONE. After careful consideration, we feel that it has merit but does not fully meet PLOS ONE’s publication criteria as it currently stands. Therefore, we invite you to submit a revised version.After careful review, there are a certain number of modifications required, as listed below.

We look forward to receiving your revised manuscript.

Kind regards,

Adrian J.F. Luty, PhD

Academic Editor

PLOS ONE

Journal Requirements:

Additional Editor Comments (if provided):

Modification of the manuscript according to the following comments will help improve its overall quality:

1. MDA regimens: lines 117 - 121 indicate that, since 2018, school-aged children (SAC) were the sole recipients of (annual) ivermectin plus albendazole, although this age-group had been receiving albendazole every 6 months up until 2016 when it was deemed no longer necessary. Can the authors clarify from when SAC received albendazole every 6 months, and indicate how that treatment was integrated with the combined ivermectin-albendazole MDA given to all aged >5 up until 2017. Also, in the Sierra Leonean context, for clarity they should specify what age-group constitutes SAC.

2. Sample procedures: lines 153 - 159 outline the quality control procedure used for microscopical diagnosis of plasmodial parasites. Was no such confirmatory procedure adopted for the Kato-Katz-based diagnoses?

3. The rate of agreement to participate by household heads was appreciably lower in the non-riverine community. If there were consistent reasons found for this, they should be detailed.

4. The authors should clarify whether the Ebola vaccines under study are intended to be tested in specific age groups.

Reviewers' comments:

Reviewer's Responses to Questions

**Comments to the Author**

1. Is the manuscript technically sound, and do the data support the conclusions?

Reviewer #1: Yes

2. Has the statistical analysis been performed appropriately and rigorously? 

Reviewer #1: Yes

3. Have the authors made all data underlying the findings in their manuscript fully available?

Reviewer #1: Yes

4. Is the manuscript presented in an intelligible fashion and written in standard English?

Reviewer #1: Yes

5. Review Comments to the Author

Reviewer #1: The authors present their manuscript on the prevalence of malaria and helminth infections in rural communities in northern Sierra Leone. This is a baseline study to inform Ebola vaccine study protocols. The results are great to present in the manuscript, however the presentation of the results could be improved to bring clarity to the manuscript. The following are my specific comments:

Abstract section

In the results section of the abstract, the authors could include at least brief results of the lymphatic filariasis and onchocerciasis results.

Results section

- The headings in the results section should be improved to bring understanding to what is being presented. The first item presented is demography stuff so the authors could have a heading like “Demographic characteristics of the study population”. Currently there is no heading for this presentation.

Again such headings as “Helminths” does not tell much. It could have been “Helminth infections in the population” etc

- Under Malaria parasitaemia, the authors could present the co-infection parasites eg how many patients were co-infected with Plasmodium falciparum and P. vivax and all the other coinfections too. The scientific world will be interested to know which parasites infected the people as coinfection.

- Also with the helminth infections, the authors do not state which parasites were the co-infections. Such information are important to publish.

- It would have been great if the authors present the different parasites differently under different groups. For instance the presentation could have been done under the following:, Soil Transmitted helminths, Schistosomiasis, blood associated parasites (Onchocerciasis and Wuchereria bancrofti). This would have brought clarity to the results.

6. PLOS authors have the option to publish the peer review history of their article (what does this mean?). If published, this will include your full peer review and any attached files.

Reviewer #1: No

---

## [Author Response · Author response to Decision Letter 0]

4 Jun 2022

Reviewers’ comments and actions taken.

We are grateful to the reviewers for their compliments and critical review of the manuscript. We have found their comments to be useful and we have accordingly revised the manuscript. 

Below is a point-by-point account of how we have used the comments to revise the manuscript.

Comment 1

1. MDA regimens: lines 117 - 121 indicate that, since 2018, school-aged children (SAC) were the sole recipients of (annual) ivermectin plus albendazole, although this age-group had been receiving albendazole every 6 months up until 2016 when it was deemed no longer necessary. 

Can the authors clarify from when SAC received albendazole every 6 months, and indicate how that treatment was integrated with the combined ivermectin-albendazole MDA given to all aged >5 up until 2017. Also, in the Sierra Leonean context, for clarity they should specify what age-group constitutes SAC.

Response 1

We have revised the manuscript in lines 122-128 with a clarification of this section as follows

Deworming of School-Aged Children (SAC) with albendazole has been part of the public health system in Sierra Leone for many years. The program defined SAC to be children aged between 5 and 14 years. A national impact survey in 2016 however demonstrated that the intervention was no longer justified. MDA-LF with ivermectin and with albendazole was undertaken annually from 2008-2017 for everyone over 5 years of age. Since 2018 however, this activity has been targeted at only SAC. Deworming of pre-SAC with albendazole continues to be undertaken every six months. This is done through local health facilities and community outreach services.

.

Comment 2

2. Sample procedures: lines 153 - 159 outline the quality control procedure used for microscopical diagnosis of plasmodial parasites. Was no such confirmatory procedure adopted for the Kato-Katz-based diagnoses?

Response 2

We have revised the manuscript in lines 177-179 to improve clarity as follows:

As part of quality control for the procedure, the microscopic examination took place within 30 minutes after sample processing. Ten percent of negative samples and all positive samples were reread by a senior laboratory staff immediately after the first reading to validate the results.

Comment 3

3. The rate of agreement to participate by household heads was appreciably lower in the non-riverine community. If there were consistent reasons found for this, they should be detailed.

Response 3

We agree to the suggestion and have revised in lines 307-311 to include the following

Response to participation in the survey was better in the riverine than non-riverine communities. The riverine communities are closer to the offices of the Ebola vaccine project. It is conceivable that community engagements activities in the Ebola vaccine project may have impacted positively on the willingness of people in the riverine communities to participate in the survey compared to people in the non-riverine communities. 

Comment 4

4. The authors should clarify whether the Ebola vaccines under study are intended to be tested in specific age groups.

Response 4

We have revised the manuscript in lines 86-87 as follows

The Ebola vaccines are evaluated in persons aged one year and older. 

Reviewer #1: 

Comment 5

The authors present their manuscript on the prevalence of malaria and helminth infections in rural communities in northern Sierra Leone. This is a baseline study to inform Ebola vaccine study protocols. The results are great to present in the manuscript; however, the presentation of the results could be improved to bring clarity to the manuscript. The following are my specific comments:

Abstract section: In the results section of the abstract, the authors could include at least brief results of the lymphatic filariasis and onchocerciasis results.

Response 5

We agree with the reviewer. Within the constraints of word limitation, we have included the following statement in the Results section (lines 58-59) of the abstract

Onchocerca volvulus and Wuchereria bancrofti infections were found in 3.3% and 0.4% of participants respectively.

Comments 6

Results section

- The headings in the results section should be improved to bring understanding to what is being presented. The first item presented is demography stuff so the authors could have a heading like “Demographic characteristics of the study population”. Currently there is no heading for this presentation. Again such headings as “Helminths” does not tell much. It could have been “Helminth infections in the population” etc

It would have been great if the authors present the different parasites differently under different groups. For instance, the presentation could have been done under the following: Soil Transmitted helminths, Schistosomiasis, blood associated parasites (Onchocerciasis and Wuchereria bancrofti). This would have brought clarity to the results.

Response 6

We appreciate the point made by the reviewer. We are however further guided by the “Instructions to Authors” provided by PLOS to keep headings and sub-headings to a minimum. 

Taking the reviewer’s suggestions into account and recognizing the limitations imposed by the journal’s requirements, we have revised the manuscript to includes the following sub-headings:

• Soil-transmitted helminths and schistosome infections (line 265)

• Blood associated parasites (O. volvulus and W. bancrofti) (line 292)

We believe this has improved the clarity of presentation of the results.

Comment 7

Under Malaria parasitaemia, the authors could present the co-infection parasites eg how many patients were co-infected with Plasmodium falciparum and P. vivax and all the other coinfections too. The scientific world will be interested to know which parasites infected the people as coinfection.

Response 7

The manuscript has been revised in lines 241-247 to include this information as follows.

A single malaria parasite species was identified in 288 slides, two species were identified in 13 slides (11 involved mixed infections of P. falciparum and P. malariae while two involved mixed infections of P. falciparum and P. ovale) and speciation was not reported for 8 slides.

Comment 8

Also with the helminth infections, the authors do not state which parasites were the co-infections. Such information are important to publish.

Response 8

This information is provided in the manuscript (lines 267-271) as follows

The positive helminth infections comprised 102 (12.5%) cases of hookworm, 11 (1.3%) cases of T. trichiura, 10 (1.2%) cases of S. mansoni and two (0.2%) cases of A. lumbricoides. The three stool samples showing mixed infections were two samples with hookworm and T. trichiura mixed infection, and one with hookworm and schistosomiasis mixed infection (Table 2).

 

---

## [Editor Report · Decision Letter 1]

22 Jun 2022

Prevalence of malaria and helminth infections in rural communities in northern Sierra Leone, a baseline study to inform Ebola vaccine study protocols

PONE-D-21-38636R1

Dear Dr. Baiden,

We’re pleased to inform you that your manuscript has been judged scientifically suitable for publication and will be formally accepted for publication once it meets all outstanding technical requirements.

Kind regards,

Adrian J.F. Luty, PhD

Academic Editor

PLOS ONE
---

## [Editor Report · Acceptance letter]

27 Jun 2022

PONE-D-21-38636R1 

Prevalence of malaria and helminth infections in rural communities in northern Sierra Leone, a baseline study to inform Ebola vaccine study protocols 

Dear Dr. Baiden:

I'm pleased to inform you that your manuscript has been deemed suitable for publication in PLOS ONE. Congratulations! Your manuscript is now with our production department. 

Kind regards, 

on behalf of

Dr. Adrian J.F. Luty 

Academic Editor

PLOS ONE